# Discriminant Analysis PCA-LDA Assisted Surface-Enhanced Raman Spectroscopy for Direct Identification of Malaria-Infected Red Blood Cells

**DOI:** 10.3390/mps5030049

**Published:** 2022-06-10

**Authors:** Gunganist Kongklad, Ratchapak Chitaree, Tana Taechalertpaisarn, Nathinee Panvisavas, Noppadon Nuntawong

**Affiliations:** 1Department of Physics, Faculty of Science, Mahidol University, Bangkok 10400, Thailand; gunganist.kog@student.mahidol.ac.th; 2Department of Microbiology, Faculty of Science, Mahidol University, Bangkok 10400, Thailand; tana.tac@mahidol.ac.th; 3Department of Plant, Faculty of Science, Mahidol University, Bangkok 10400, Thailand; nathinee.pan@mahidol.ac.th; 4National Electronics and Computer Technology Center (NECTEC), 112 Thailand Science Park, Pathum Thani 12120, Thailand; noppadon.nuntawong@nectec.or.th

**Keywords:** malaria-infected red blood cells, *P. falciparum* (3D7), surfaced-enhanced Raman spectra, PCA-LDA

## Abstract

Various methods for detecting malaria have been developed in recent years, each with its own set of advantages. These methods include microscopic, antigen-based, and molecular-based analysis of blood samples. This study aimed to develop a new, alternative procedure for clinical use by using a large data set of surface-enhanced Raman spectra to distinguish normal and infected red blood cells. PCA-LDA algorithms were used to produce models for separating *P. falciparum* (3D7)-infected red blood cells and normal red blood cells based on their Raman spectra. Both average normalized spectra and spectral imaging were considered. However, these initial spectra could hardly differentiate normal cells from the infected cells. Then, discrimination analysis was applied to assist in the classification and visualization of the different spectral data sets. The results showed a clear separation in the PCA-LDA coordinate. A blind test was also carried out to evaluate the efficiency of the PCA-LDA separation model and achieved a prediction accuracy of up to 80%. Considering that the PCA-LDA separation accuracy will improve when a larger set of training data is incorporated into the existing database, the proposed method could be highly effective for the identification of malaria-infected red blood cells.

## 1. Introduction

Malaria is a disease that is transmitted by mosquitoes (acting as a vector) and it is caused by a parasite of the *Plasmodium* genus. *Plasmodium falciparum* is usually the source of malaria mortality [1]. Between 2000 and 2020, the World Health Organization estimated that more than 200 million cases of malaria infection were reported each year [2]. The gold standard techniques for malaria diagnosis are rapid diagnostic tests (RDTs) and microscopic analysis of blood-smears [3,4]. The blood smear test is highly sensitive and precise for malaria diagnosis, but it needs to be performed by trained and qualified staff. RDTs are rapid; however, their sensitivity is not comparable to blood smear examination. Another highly sensitive diagnostic approach is polymerase chain reaction (PCR) but it involves a lengthy sample preparation process [5,6]. Various methods have been developed over the last two decades to satisfy the demand for simple and highly sensitive diagnostic techniques.

There are a number of reports on the application of the Raman spectroscopy technique in malaria diagnosis. The majority of these studies have focused on detecting hemozoin, or malaria pigment [7]. After the parasite infects an erythrocyte, the host’s hemoglobin is catabolized. A by-product is created when the free heme is converted into a crystalline structure known as “hemozoin”. Plasma or red blood cell extract is often used as the sample. Wood et al. first recorded the expression of the hemozoin Raman signal in the *Plasmodium* parasite food vacuole [8], and subsequent investigations of the excitation wavelength led to the enhancement of the Raman signal by using the resonance wavelength activated by hemoglobin and hemozoin [9,10,11,12]. In addition to the resonance Raman spectroscopy, tip-enhanced Raman scatter and magnetic fields have been used to increase the Raman intensity for hemozoin detection [13,14]. The level of hemozoin in the blood clearly corresponds to the time of infection. A small volume of hemozoin makes the examination difficult, especially in the early stages of infection [15]. Due to the size and structure of the metallic nanoscale substrate, the surface-enhanced Raman substrate was found to significantly enhance the Raman intensity [16,17]. With this technique, hemozoin in 0.0005% infected red blood cells could be detected by using a gold-coated butterfly wing nanostructure [18]. Chen et al. used a silver nanorod structure as the surface-enhanced Raman substrate to measure the modification of the red blood cell membrane and the expression of the protein component, especially the cytoadherence protein complex in the erythrocyte infection phase [19]. This was the first time that the direct measurement of the malaria-infected red blood cell was accomplished. Thereafter, there was no further work on the direct measurement of malaria-infected red blood cells. However, the Raman spectroscopy approach for malaria detection is still being developed. In 2020, gold substrate nanoparticles were employed for antibody-free malaria detection in whole blood [20]. Then, coupling Raman spectroscopy with mid- and near-infrared spectroscopy for malaria and arboviruses in human blood [21] was reported in 2021.

Apart from Raman spectrum investigation, the discrimination analysis technique is often used as an analytical tool in the Raman spectroscopy research area, particularly in regard to the issue of classification. The Raman spectrum is considered to be multivariate data and multivariate discrimination analysis (e.g., PCA and PC-LDA [21,22,23,24,25,26,27]) has been applied to differentiate a variety of cases such as the state of the red blood cell infection [19], the normal and the malaria-infected spleen tissue [28], and dengue and malaria sera Raman spectra [29].

In this work, we report for the first time that the direct identification of the malaria-infected red blood cells can be accomplished by the combination of SERS and principal component analysis coupled with linear discrimination analysis (PCA-LDA). The surface-enhanced Raman spectra were obtained from direct measurements of the normal and 3D7 *P. falciparum*-infected red blood cells deposited on a SERS substrate. A silver nanorod structure was used as the SERS substrate. The characteristic spectra and Raman spectral images of normal and infected red blood cells were presented. Subsequently, owing to the large number of spectra collected, the data set of spectra was used to train the separation model using a PCA-LDA-based machine learning algorithm. The proposed method can achieve a separation accuracy of more than 90%. Finally, the volunteers’ blind data sets were used to evaluate the model and provided a prediction accuracy of up to 80%. Importantly, the accuracy of the separation model could be continually extended due to the updateable Raman spectra data set and could have the potential to be developed for the clinical application of malaria diagnosis in the future.

## 2. Material

### 2.1. Blood

Human red blood cells from healthy volunteers prepared at the National Center for Genetic Engineering and Biotechnology (BIOTEC), Thailand were used in this study.

### 2.2. Surface-Enhanced Raman Substrate

The OnSpec chip used in this study is a SERS based on a silver nanorod structure. All substrates were prepared by NECTEC, Thailand with a laboratory-made DC magnetron sputtering system, with a glancing-angle deposition (GLAD) technique. The details of the SERS preparation and fabrication are described in [30,31].

### 2.3. Raman Spectrometer

The Renishaw inVia^TM^ confocal Raman microscope with an He-Cd laser was used. The light source provided an excitation wavelength of 532 nm. The microscope system was a Leica microsystem, which can be equipped with magnification objective lenses of 5x, 20x, 50x and 100x. The CCD camera was fitted with up to four detectors such as electron multiplied (EM) and InGaAs arrays. The Renishaw inVia^TM^ confocal Raman microscope captured the spectra with a spectral resolution of 0.3 cm^−1^ (FWHM).

### 2.4. Software

WiRE4.2 software was used for spectral processing including cosmic ray removal, baseline subtraction, smoothing signal and normalization.

The imaging processes and separation model were simulated by Python. The module Hyperspy 1.6.5 was applied to intensify the imaging of the spectra in the area of interest and a collection of machine-learning modules, including scikit-learn 0.23.2 were applied for PCA-LDA discrimination processing.

## 3. Methodology

In this work, there are 3 main parts (see Figure 1). The first part involved the blood sample preparation and then the Raman spectra were collected. Finally, the data were interpreted and used for training the separation model.

### 3.1. Sample Preparation

#### 3.1.1. Malaria-Infected Cell Sample

*P. falciparum* (3D7) parasites were cultured according to the routine method [32] in RPMI1640 medium supplemented with 1% Albumax I at 4% hematocrit. The parasite culture was maintained at 37 °C with 5% carbon dioxide. The age of the parasite was around 24–32 h post invasion (this corresponds to the mid–late trophozoite stage to the early schizont stage of the asexual cycle). The Percoll density gradients method was used to enrich the infected red blood cells. The percentage of parasite-containing red blood cells was confirmed by Giemsa staining. Red blood cells were resuspended in phosphate buffered saline (PBS) at 0.16% hematocrit. Then, an aliquot of 25 μL of the cell suspension was applied to the OnSpec chip. The sample on the surface of the OnSpec chip was flattened by the spin coater.

#### 3.1.2. Normal Red Blood Cell Sample

The normal red blood cell sample was prepared with human red blood cells resuspended in PBS. Then, 25 μL of the cell suspension was applied to the OnSpec chip in the same way as the infected sample.

### 3.2. Spectral Collection

#### 3.2.1. Measurement Conditions

All spectra were acquired in the range of 123–1883 cm^−1^ with 50x objective lens, and the laser power was set to be 1% (around 3 mW) of the maximum power (the low power of the laser was used to avoid damage to the sample). Each spectrum was presented as the average of three accumulation spectra with a 30 s exposure time. 

#### 3.2.2. Pointing Spectral Collection

Point spectral collection was applied to select the point in the substrate area. The laser pointed to the pre-selected point. Normal and infected red blood cell spectra were collected from the points in the cell area, while outside the cell, spectra were collected from the points outside the cell area. There were 460 normal red blood cell spectra acquired from 6 normal red blood cell sample sets, 365 infected red blood cell spectra acquired from 4 infected red blood cell sample sets and 350 outside-the-cell spectra acquired from 10 total sample sets. Moreover, blank sample tested only on PBS was dropped on the OnSpec chip. All spectra met the measurement conditions described in Section 3.2.1.

#### 3.2.3. Area Spectral Collection

The area spectral collection is the spectral collecting process done by area selection. In the area of interest (the red boxes in Figure 2), the intersection of the grid in x and y (see Figure 3) was the excitement point. The Raman spectrum from each position is the 2D data that contains the information about the intensity and wavenumber. The spectrum from each position was acquired using the same measurement conditions described in Section 3.2.1.

### 3.3. Data Analysis

#### 3.3.1. Spectral Processing

In the WiRE 4.2 program, there are built-in tool for spectral processing. The polynomial order was defined and adjusted automatically for fitting the baseline of the spectrum. The Savitsky–Golay filter [33] was used in the smoothing process. The cosmic ray spikes were identified by the threshold value of the width and height of the intensity peak. Then, the mean value of the intensities around the spike was used as the substitute intensity at the spike position [34,35]. Finally, the spectra were only truncated in the range covering 450–1750 cm^−1^.

#### 3.3.2. PCA-LDA Separation Model


*PCA-LDA Discrimination Analysis*


Principal component analysis (PCA) is an unsupervised technique that is frequently used in conjunction with Raman spectral analysis [36]. The Raman spectrum is a representation of multidimensional data. PCA is always used to reduce the dimensions of the data and visualize the spectrum by transforming the data set into a new coordinate frame. Apart from dimension reduction, PCA can also be used to extract data features. While reducing the dimension of the data, the information related to the data is still retained as much as possible. Principal components (PCs) were used as new orthogonal coordinates. The first PC (PC1) is responsible for the data’s largest variance axis, followed by PC2, and so on. 

After spectral processing, 1175 Raman spectra were analyzed using the PCA algorithm: 460 from normal cells, 365 from infected cells, and 350 from outside the cell area. The 350 PCs transformed data that account for >99.9 percent of the total variance was used as the LDA algorithm’s input data. LDA is the supervised analysis [37]. Each transformed spectrum from the PCA was labeled according to its cell type: normal, infected, or outside the cell. Figure 4 shows the flow chart of the discrimination analysis. All spectra acquired from the experiment were separated into a training data set (80%) and testing data set (20%). The training spectra were transformed by the PCA and LDA procedures. From the original data, each spectrum containing the information about intensities and wavenumbers (Raman shift) was transformed into data dots in 2D dimensional coordinates of LDA in which each dot represents the information about the spectrum. The PCA-LDA separation model was created by grouping the transformed data in 2D LDA coordinates according to their spectrum class. Then, the testing spectra was transformed in the same way from the original spectra to the data dots in the PCA-LDA separation model for the evaluation. Finally, all spectra classified as the training spectra and the blind spectra were applied as the testing spectra. The pie chart is the output of the blind test, which shows the probabilities of the spectra class.


*Blind Test*


Six blind samples were prepared according to the method described in Section 3.1.1 and Section 3.1.2. Three of these were normal cell samples and the others were infected cell samples. Six samples were randomly distributed for testing by three operators. The operators must maintain the spectral collection parameter in accordance with the manual. The manual defines the method for obtaining the spectra and the measurement parameters such as the laser’s power and exposure time (as defined in Section 3.2.1). Each sample contained 30–35 spectra randomly collected from the cells. Preprocessing (e.g., cosmic ray removal, baseline subtraction, normalization, and spectral range selection) was applied to the spectra from each blind spectrum. This process may result in a reduction in the number of usable spectra. The blind spectra of each sample were then transformed into PCA and LDA coordinates.

Besides, two more spectra sets were collected from outside the cell using the same protocols. Without a doubt, the spectra could not be considered as a blind sample due to the point spectral collection process. These two data sets were used exclusively to evaluate the PCA-LDA separation model. However, the outside-the-cell spectra database may be useful in future analysis when area spectral collection is used. Consequently, there were eight data sets in total for the PCA-LDA separation model evaluation.

## 4. Results and Discussion

### 4.1. Characterization of Normal and Infected Red Blood Cell Raman Spectra

After the spectral processing, the average spectrum is shown by the blue line in Figure 5a, representing the average of 460 spectra of normal red blood cells, while 365 *P. falciparum* (3D7) infected red blood cell spectra were averaged and are represented by the red line in Figure 5a. Additionally, the spectrum for PBS used as the cell’s media is displayed as the green line in Figure 5a. The Raman normalized intensity peak characteristics of normal and infected red blood cells are shown together in Figure 5b.

When normalized to the intensity at 1586 cm^−1^, the intensity signal of the Raman spectra from normal and infected red blood cells shows a slight difference, as seen in Figure 5b. This is due to the fact that the intensity at Raman shift 1586 cm^−1^ is the highest peak in all the spectra acquired from the cell area.

For the intensity imaging, the peaks were focused at 747, 1128, 1228, 1372, 1560 and 1620. Therefore, the normalized intensities in the ranges of 747–749, 1124–1134, 1220–1237, 1361–1383, 1555–1565, and 1615–1629 cm^−1^ (the wavenumber ranges that cover the peaks of interest) were averaged. The variation in the intensity is represented by the color of the image as shown in Figure 2.

The disposal product, hemozoin is primarily visible between 24 and 32 h after infection, corresponding to the mid–late trophozoite to schizont stages. The hemozoin and the mechanism of its formation are of great interest in the development of anti-malarial drugs. Throughout the parasite’s life cycle, hemoglobin in the red blood cell is catabolized, producing a large amount of free heme that is toxic to the parasite. Hemozoin is a byproduct of the parasite’s free heme detoxification mechanism [38]. The components’ peaks in the obtained Raman spectrum were assigned according to Table 1. Hemoglobin is the primary constituent of normal red blood cells and its presence is represented by the strong band in the Raman spectrum resulting from a 532 nm laser excitation wavelength to normal and infected red blood cells. However, hemoglobin and hemozoin both contain the heme prosthetic group, which provides the resonance enhancement in this wavelength range of excitation [11,28,39]. The Raman spectra of hemoglobin and hemozoin were comparable with 532 excitation wavelengths [28]. Their distinctive peaks are in close proximity to one another as reported by Frame et al. [28]. The experiments yielded the corresponding results via the characteristic of the averaged normalized Raman spectrum and imaging (in Figure 3 and Figure 5b). The obvious difference between the normal and infected cells is visible at Raman shifts ~1620, 1560, 1372, 1228,1128, and 747 cm^−1^. When red blood cells are infected, the hemoglobin was converted to the Fe^3+^ structure (hemozoin) [7,8,11,40]. The normalized Raman spectrum exhibits an increasing intensity trend of Raman shifts at ~1620, 1560, 1372 and 1228 cm^−1^, indicating the Fe^3+^ hemozoin characteristic band, while the Raman shift at ~747 and ~1128 cm^−1^, which predominantly presents in the hemoglobin band exhibits the opposite trend.

### 4.2. Discrimination Analysis

Due to the slight differences between the spectra, it is difficult to distinguish the Raman spectra of normal cells from that of infected cells with the naked eye. Moreover, several hours must be spent on an area spectra collection for imaging. As a result, PCA and LDA techniques [50,51,52] were chosen for the discrimination analysis in this work.

The loadings plot of PC1 and PC2 are shown in Figure 6b, while the spectra in the range of 450–1750 cm^−1^ were reduced in dimension from 755 to 350 and transformed to a visual representation in 2D as shown in Figure 6c (PC1 and PC2). The scattering plot allows the separation of cell spectra (both normal and infected cell spectra) from non-cell spectra (spectra collected from outside the cell area). However, the normal and infected spectra was still misclassified. PC1 represented 69.1% of the variance, while PC2 represented 7.8%. There were 755 dimensions of data in total; nearly all significant features were extracted using 350PCs, and these 350PCs data were then used as the input for the LDA algorithm in a subsequent step.

The LDA algorithm was used to process 1175 spectra. The discrimination of each type of data is demonstrated clearly in the LD1 and LD2 coordinates in Figure 6d. The pale sky-blue area represents a collection of normal cell spectra. The light-red cluster represents the infected cell, while the light-green area represents the non-cell spectra. The distinguishing area of each spectral class in Figure 6d was used as criteria for predicting the type of unknown data. 

To illustrate the analytical parameters that indicated the PCA-LDA separation model’s efficiency, 1175 spectra were split into an 80% training set and a 20% preliminary testing set for the model evaluation. In data splitting, ten random states (0–9) were selected. Thus, ten data sets were considered for the model evaluation, and the final result shown in Table 2 was calculated using the average value of the confusion matrix. The values in the confusion matrix were used to calculate the accuracy, precision, sensitivity (recall), f1-score, and specificity terms for the PCA-LDA separation model, as shown in Figure 7. Additionally, the ROC curve provided an overview of the efficiency of the PCA-LDA separation. If the area under the receiver operating characteristic curve (ROC) is close to a value of one the classification is considered to be well-performed. Due to the fact that this study contained data from three distinct classes, the data was binarized prior to the creation of the ROC curve. The ROC curve in Figure 8 illustrates that the PCA-LDA model performed well in terms of discrimination, with an area under the curve of more than 0.9 for all classes’ analysis.

### 4.3. Blind Test

An example of the blind results is depicted in Figure 9 and is represented by the yellow dots. At the conclusion of the procedure, the pie chart was used to represent the data class prediction. The position of unknown data in the PCA-LDA separation was converted to the probability of data class prediction, which could be easily visualized using a pie chart. The area of the pie chart indicates the sample class’s probability (see Figure 10).

After predicting the type of blind samples using the PCA-LDA separation model generated from 1175 training spectra, the unknown data class prediction results were displayed in the pie chart (see the example in Figure 10, all pie charts of the blind data sets can be seen in the Appendix A). The summary of all sample predictions and their expected outcome are shown in Table 3. All blind sample classes could be accurately predicted. Each prediction class corresponded to an expected class.

Table 4 shows the accuracy, precision, sensitivity (recall), and specificity of the blind data sets. The values dropped compared with using 20% split data from 1175 training spectra to be tested (see Table 2). In Figure 11, the area under the receiver operating characteristic curve shows the same trend, with the infected class having an area under the curve of 0.78 and the normal class having an area under the curve of 0.80. However, the efficiency of the out-of-cell class is quite clearly stable, regardless of whether the model was evaluated using blind data or data split from the 1175 training spectra set. That is, the separation model is highly efficient at classifying the inside- and outside-the-cell information. 

In Figure 8, the testing set taken from the 1175 training spectra, the model demonstrates a high ability to discriminate with an average accuracy of over 90%, as illustrated in Table 2, and the area under the receiver operating characteristic curve is nearly one (Figure 8). It should be noted that in blind spectra data sets obtained from various operators, the distribution that was generated in each sample’s spectral data varied, despite the fact that measurement parameters such as the laser’s power, the time exposed to the sample, the lens, and so on were all controlled, including the procedures used in the sample preparation step. As the distribution in the training data was incompatible with the distribution in the testing data set, the efficacy of discriminating between data classes was reduced. However, the training spectra set in the PCA-LDA separation model is an updateable database and it can be updated infinitely. When a larger set of training data that covers the distribution in the testing set is incorporated into the existing database, the PCA-LDA separation could improve the efficiency of malaria-infected red blood cells identification.

## 5. Conclusions

Nowadays, there is still demand for the development of malaria diagnosis techniques. This study aims to develop an alternative method for malaria detection based on surface-enhanced Raman spectroscopy. Three crucial stages including the sample preparation, collecting the Raman spectra and data analysis are described in detail. Both average normalized spectra and spectral imaging were found to produce comparable results. During infection, the Raman intensity modification of the heme-based composition, which corresponds to the metabolically crystallized byproduct (hemozoin) of hemoglobin digestion, increased at ~1620, 1560, 1372, and 1228 cm^−1^, and decreased at 747 and 1128 cm^−1^. When PCA-LDA analysis was applied to the SERS spectra, the separation of infected and normal cell spectra was readily apparent in PCA-LDA coordinates. Additionally, the PCA-LDA coordinate separation could be used to predict the class of the blind sample cell. Each data set yielded the correct response. The separation model’s efficiency can still be improved since the spectral database for training the model can be updated. In this study, only the qualitative results of the sample classification were reported. However, the protocols that were applied can be reproduced by volunteers and all of the blind samples offered accurate predictions. This shows that the proposed approach can be employed in future quantitative malaria diagnosis development studies and can be refined so that it can be applied at a clinical application level. Furthermore, the spectra in this study can be utilized as references for any red blood cell or malaria-related Raman spectroscopy investigation.

## Figures and Tables

**Figure 1 mps-05-00049-f001:**
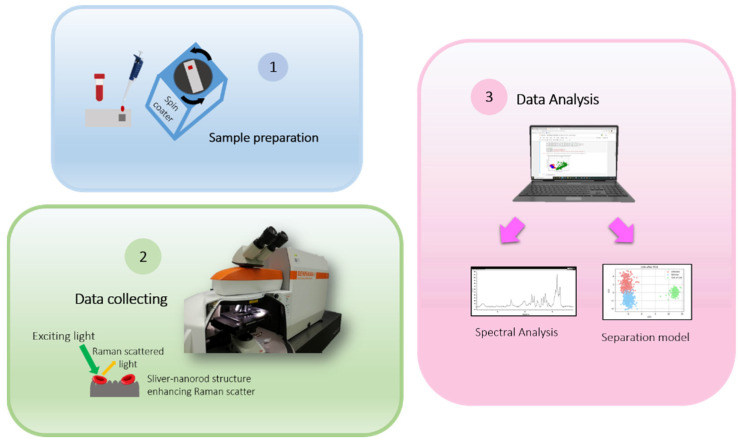
Diagram of the Methodology.

**Figure 2 mps-05-00049-f002:**
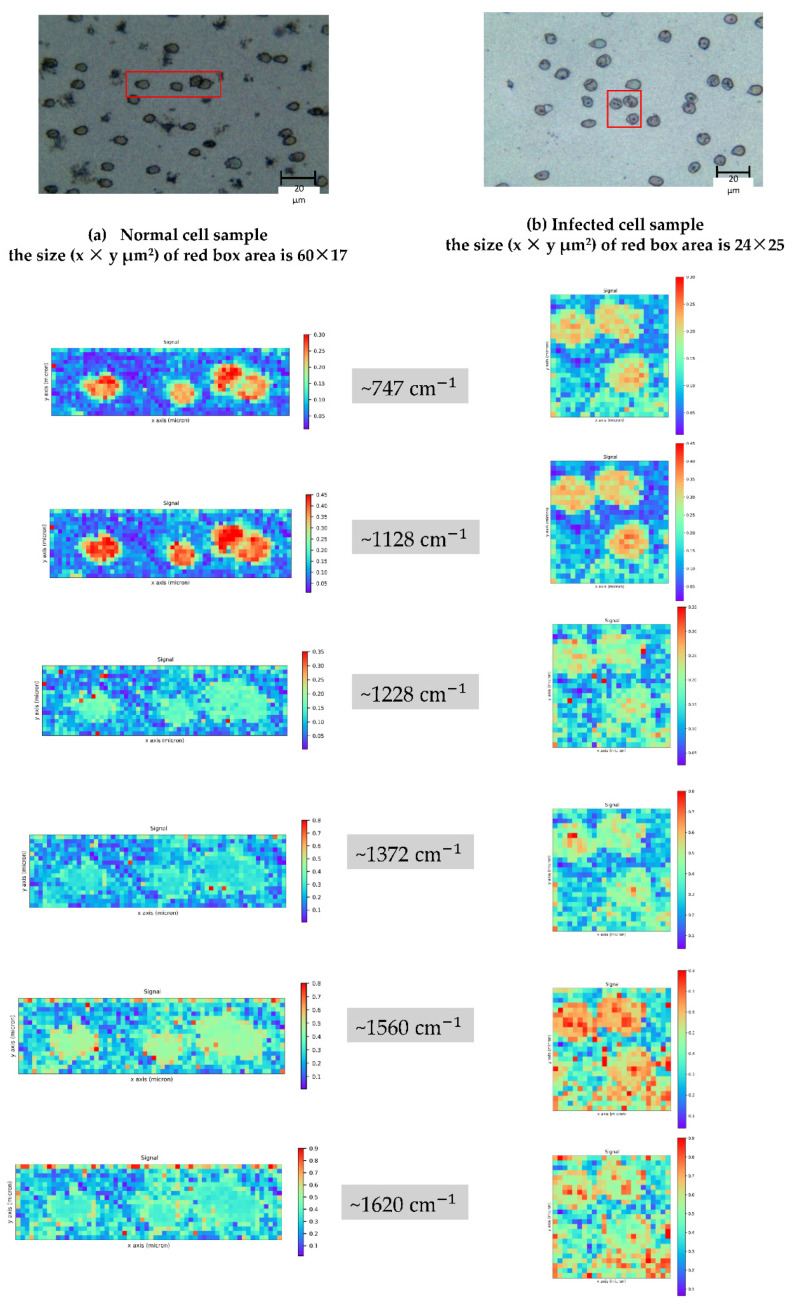
Image shows white light images of normal (**a**) and infected red blood cells. (**b**) The red boxes indicate the mapped area with the normalized intensities covering the peaks at 747, 1128, 1228, 1372, 1560 and 1620 cm^−1^. These maps visually compare the healthy red blood cells and the *P. falciparum* (3D7) infected red blood cells. The increase or decrease in the intensity of the Raman spectrum can be linked with the biological component’s modification in the red blood cell. The information regarding peak assignment is shown in Table 1.

**Figure 3 mps-05-00049-f003:**
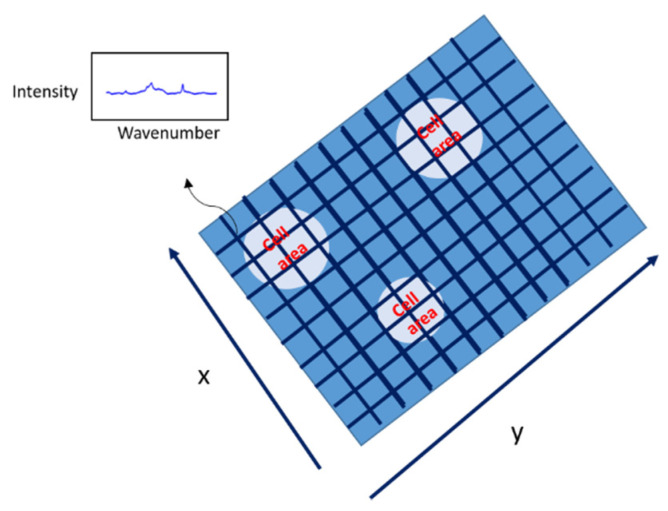
Each point at the intersection in the area provided a Raman spectrum. The Raman spectra and spatial data were used for intensity image simulation.

**Figure 4 mps-05-00049-f004:**
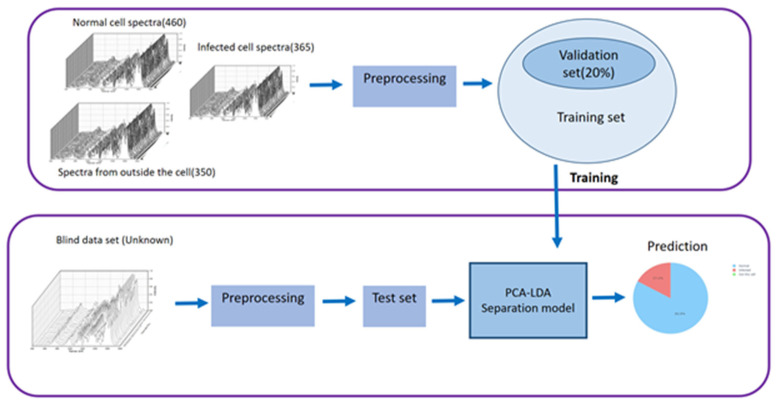
Flow chart of the discrimination analysis.

**Figure 5 mps-05-00049-f005:**
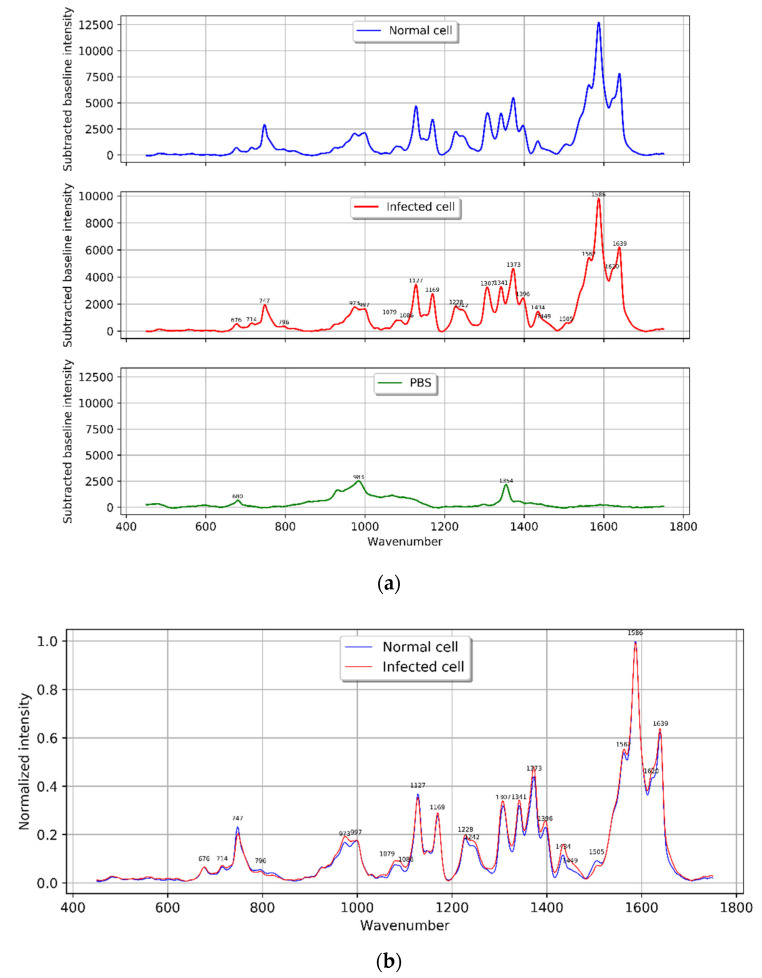
Analysis of Raman spectrum intensity for normal and *P. falciparum* (3D7)-infected red blood cell. (**a**) Averaged normal red blood cells spectra (*n* = 460) and infected red blood cells spectra (*n* = 365) in comparison to PBS buffer. (**b**) Averaged, normalized spectra from normal red blood cells (*n* = 460) and infected red blood cells spectra (*n* = 365).

**Figure 6 mps-05-00049-f006:**
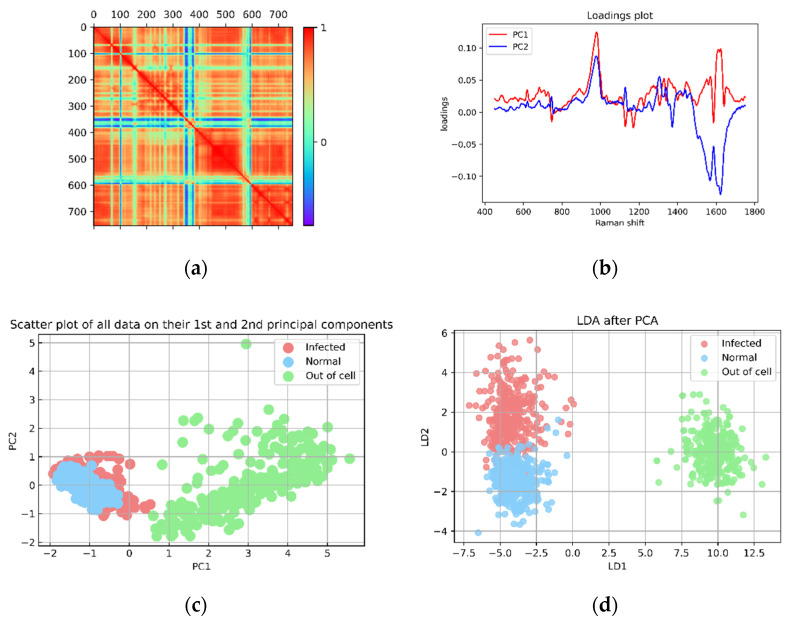
PCA-LDA results. (**a**) The covariance matrix of the PCA plot. (**b**) The loading plot of PC1 and PC2 after PCA processing of 1175 Raman spectra (460 normal cells, 365 infected cells and 350 outside-the-cell spectra) by using 755 Raman shifts in the range of 450–1750 cm^−1^ is presented. (**c**) The PCA result shows the PC1 and PC2 coordinate and 350 PC results from PCA are continually input to LDA processing. (**d**) The LDA result plots in LD1 and LD2. In (**c**,**d**): blue dots are designated as the normal cell transformed spectra, red dots represent the infected cell transformed spectra and green dots are the transformed spectra obtained from the outside-cell area.

**Figure 7 mps-05-00049-f007:**
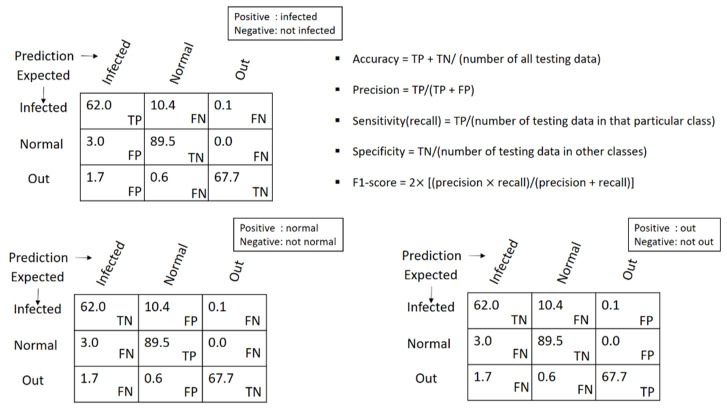
Confusion matrix and each parameter calculation.

**Figure 8 mps-05-00049-f008:**
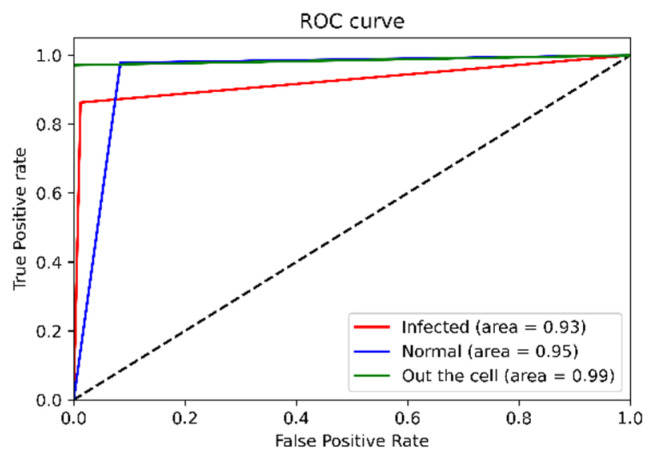
ROC curve presenting the efficiency of the model evaluated by 20% split data from training data set.

**Figure 9 mps-05-00049-f009:**
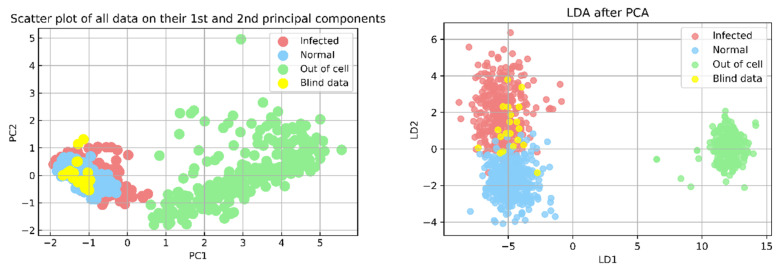
Transformation of blind data to PCA-LDA coordinates.

**Figure 10 mps-05-00049-f010:**
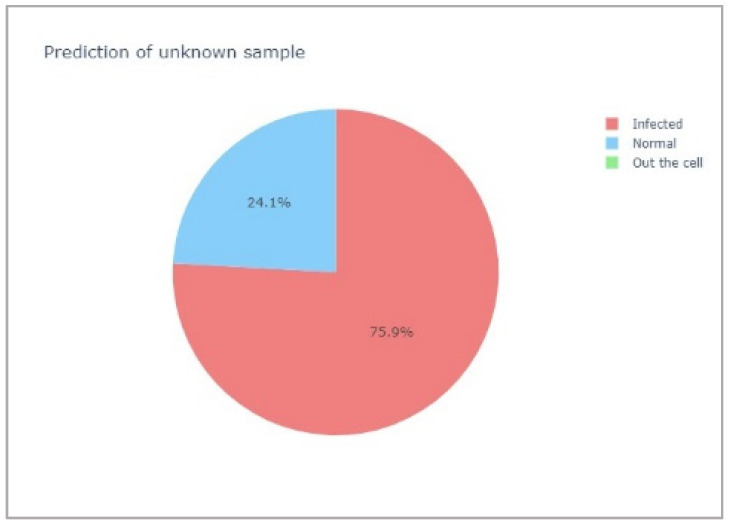
An example of a prediction pie chart.

**Figure 11 mps-05-00049-f011:**
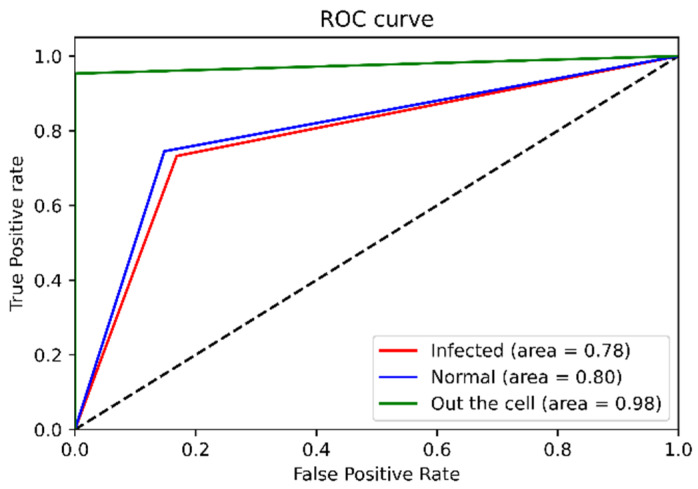
ROC curve of PCA-LDA model evaluation using blind sample data sets (test set).

**Table 1 mps-05-00049-t001:** Raman peak assignment.

Observed Wavenumber (cm^−1^)	Assignment
676	υ_7_, δ(pyr deform) _sym_ of hemoglobin
747	υ15 Hb [υ(pyr breathing), Trp, O-P-O sym Str. (lipid)
796	$\upsilon \left(pyr\;breathing\right)\upsilon_{6}$
826	Porphyrin: $\gamma \left(C_{m}H\right)$, Tyr
923	Amino acids: C–COO^−^ stretch
950	C-C Str
973	υ_46_, δ(pyr deform) _asym_
997	υ_47_, υ$(C_{b}C_{1}{)}_{asym}$, Protein (Phe)
1078, 1079	$\delta {\left(=C_{b}H_{2}\right)}_{4}$, C-O Str.
1127, 1128	υ_5_
1171, 1172	υ_30_ υ(Pyr half-ring)_asym_
1225	$\delta \left(C_{m}H\right)$
1228	$\delta \left(C_{m}H\right)$ (Oxy)
1244, 1242	Amide III
1247	Amide III (collagen assignment)
1307	$\upsilon_{21}$
1340	$\upsilon_{44}$(Pyr half-ring)_sym_
1366, 1372	$\upsilon_{4}$(Pyr half-ring)_sym_
1396	$\upsilon_{20}$
1432, 1434	$\upsilon_{28}$
1440	C-H_2_ and C-H_3_ bend (protein, lipid)
1463	$\upsilon_{3}$, C-H_2_ and C-H_3_ bend
1540	$\upsilon_{11}$
1560, 1562	υ(c=c),Trp
1586	$\upsilon_{37}$
1620	$\upsilon_{\left(c=c\right)}$
1639, 1640	$\upsilon_{10}$

* References for the assignment of Raman peaks: [11,19,28,38,39,41,42,43,44,45,46,47,48,49].

**Table 2 mps-05-00049-t002:** List of parameters presenting how well the infected cells were distinguished from normal cells and non-cell area spectra (evaluated by 20% split data from training data set).

	Precision	Sensitivity (Recall)	F1-Score	Specificity	Support
Infected	0.97	0.86	0.91	0.98	73
Normal	0.88	0.98	0.93	0.92	92
Out the cell	1.00	0.97	0.99	0.93	70
Accuracy	0.94				235
Macro avg	0.95	0.94	0.94		235
Weight avg	0.94	0.94	0.94		235

**Table 3 mps-05-00049-t003:** Prediction results of blind samples.

Sample No.	Expected	Probability of Sample Class			Prediction
		Normal (%)	Infected (%)	Out (%)	
1	Infected	24.1	75.9	0.0	Infected
2	Normal	60.6	39.4	0.0	Normal
3	Infected	31.8	68.2	0.0	Infected
4	Infected	25.0	75.0	0.0	Infected
5	Normal	82.9	17.1	0.0	Normal
6	Normal	78.6	21.4	0.0	Normal
7	Out	0.0	6.2	93.8	Out
8	Out	3.1	0.0	96.9	Out

**Table 4 mps-05-00049-t004:** Efficiency of the model evaluated by blind sample data sets (test set).

	Precision	Sensitivity (Recall)	F1-Score	Specificity	Support
Infected	0.65	0.73	0.69	0.83	71
Normal	0.79	0.75	0.77	0.84	102
Out the cell	1.00	0.95	0.98	0.74	64
Accuracy	0.80				237
Macro avg	0.81	0.81	0.81		237
Weight avg	0.81	0.80	0.80		237

## Data Availability

Not applicable.

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
