# Peer review of "Discriminant Analysis PCA-LDA Assisted Surface-Enhanced Raman Spectroscopy for Direct Identification of Malaria-Infected Red Blood Cells"

_mps, 2022, doi:10.3390/mps5030049_

Round 1

Reviewer 1 Report

The authors mention how Raman spectroscopy has been explored for malaria diagnosis and the novelty of their research that is using surface enhanced Raman Spectroscopy and PCA-LDA for detection of infected red blood cells.

The authors did Raman spectra to healthy and infected red blood cell (RBC). Is not clear how normalization is performed to detect differences. Authors suggest that the differences observed is due to alterations in hemoglobin and hemozoin but it was not clear to me how they determine the specificity for this two blood components and not others, such as for example biliverdin or bilirubin (https://doi.org/10.1074/jbc.M112.414078). Can the authors say that the method explored is then based on hemozoin detection?

Another comments regards to the quantity of parasites vs quantity of hemozoin that can be important for the diagnosis of malaria. Authors mention that separation and concentration of P. falciparum is done with percoll but no information of parasitemias is given. Will be interesting also to test if differences can be detected between P. falciparum and P. vivax as mentioned in the line 78.

Another important point in the development of the technology for diagnostic of malaria, is that being able not only to qualitatively detect but quantify, the technology could have a significant impact of the follow up of parasitemia given the present treatment the artemisinin-based combination therapy. An opinion paper refers to Raman spectroscopy to potentiality going beyond detection and detail some characteristics that could take towards personalized malaria medicine (DOI: 10.1186/s12936-020-3149-4). Discussion should also rise this possibility considering the data presented.

The authors must revise the English of the manuscript. Also, the structure of the manuscript, including lines space and formatting should be adjusted.

Other minor comments:

The authors should consider change the order of fig 7 and 8, since figure 8 is firstly mentioned in the text.

Line 35: suggest to remove the sentence “Malaria is one of the diseases that mosquito are the vectors.” as is not informative for the manuscript message or alternatively replace “that mosquito” by “in which mosquitoes”.

Line 36: replace “often” by “usually”, since P. falciparum is the strain responsible for most of malaria complicated cases and death

Line 47: The sentence “There were reports on the application of the Raman spectroscopy technique in malaria diagnosis. The majority of the studies focused on detecting hemozoin, or malaria pigment.” needs reference. There is a considerable amount of reports based on hemozoin detection. I suggest a recent review article on that (DOI: 10.1021/acssensors.1c01750).

Line 64: especially? Re-phrase

Line 66: replace “This was the first accomplishment that direct measurement to the malaria-infected red blood cell accomplished” by “This was the first time that the direct measurement of the malaria-infected red blood cell was accomplished”.

Line 70: “were employed” instead of “employed”

Line 78: “the P. falciparum and P. vivax species hemozoin extraction” re-phrase

Line 81: replace “identifications” by “identification”

Line 84: Maybe it is worth to mention the SERS substrate.

Line 90: misspelling “Importanatly”

Line 100: It is not the first time that SERS is mentioned. Write the meaning of the acronym in the first time that is mentioned in the text.

Line 111: What is FWHM?

Line 112: This paragraph presents a larger spacing than the previous paragraphs. Re-adjust.

Line 115: processes or processing? Remove the “:”.

Line 116: “to intensity imaging of spectra” replace by “to intensify the imaging of the spectra”

Line 117: remove “:” and add “including”

Line 134: the authors use RBCs but never state what RBCs are. The term “red blood cells” was used several times in the text.

Line 139: replace “bllod” by “blood”; RBCs is again mentioned. Re-phrase the paragraph.

Line 144: replace “are acquired” by “were acquired”. The authors stated the methods in the past, not present.

Line 145: the authors mention that the laser power was set to be 1% from the maximum power. Why is this relevant? Add in the manuscript.

Line 149: replace “applied” by “was applied”. What point?

Line 181: remove “the”

Line 182: something is missing there. Re-phrase

Line 200/201: The authors mention for the first time the meaning of PCA. PCA and LDA were used multiple times previously in the text.

Line 251: PBS was mentioned before.

Line 281: The authors should normalize the information: either say normal or healthy red blood cells.

Line 331: what is PC1 and PC2? Are the spectra after PCA? The authors could explain better in the text.

Author Response

Dear reviewer1

We would like to express our gratitude for the opportunity to revise the above-mentioned manuscript, and sincerely appreciate the reviewers’ suggestions and comments. We appreciate your kind assistance in improving our manuscript. We tried to comply with the reviewer’s comments to the best of our ability. We hope that these revisions ensure that the paper now meets the requirements of a contributed paper for your prestigious journal.

Best Regards,

Authors

Reviewer 2 Report

The manuscript introduced one method “Discriminant analysis PCA-LDA assisted surface-enhanced Ra- 2 man spectroscopy for direct identification of malaria-infected 3 red blood cells”. the method seems to be interesting and has potential to be applied into clinical test. Before be accepted for publication. The authors should answer the following questions:

1.      Figure 4 introduced averaged, normalized spectra from normal 283 red blood cells (n=460) and infected red blood cells spectra (n=365).it is difficult to identify their difference. I think their difference is not strong enough to be used to discriminate normal red cells and infected red cells.Please offer a detail explanation.

2.      In fact,this method’s false postitive is very high. According to table 3. Prediction results of blind samples is not high. False positive rate in infected,normal,infected and infected groups  is 24.1%,39.4%31.8% and 25%,respectively ,which suggest this method is mature and have a long road to improve its precision rate.

3.      The authors should improve manuscript’s level.

Author Response

Dear reviewer2

We would like to express our gratitude for the opportunity to revise the above-mentioned manuscript, and sincerely appreciate the reviewers’ suggestions and comments. We appreciate your kind assistance in improving our manuscript. We tried to comply with the reviewer’s comments to the best of our ability. We hope that these revisions ensure that the paper now meets the requirements of a contributed paper for your prestigious journal.

Best Regards,

Authors

This manuscript is a resubmission of an earlier submission. The following is a list of the peer review reports and author responses from that submission.